# Effect of Surface Topography Parameters on Friction and Wear of Random Rough Surface

**DOI:** 10.3390/ma12172762

**Published:** 2019-08-28

**Authors:** Ruimin Shi, Bukang Wang, Zhiwei Yan, Zongyan Wang, Lei Dong

**Affiliations:** 1School of Mechanical Engineering, North University of China, Taiyuan 030051, China; 2National Engineering Laboratory of Coal Mining Machinery Equipment, Taiyuan 030051, China

**Keywords:** surface topography, dynamic pressure, friction, contact

## Abstract

In order to explore the relationship between the surface topography parameters and friction properties of a rough contact interface under fluid dynamic pressure lubrication conditions, friction experiments were carried out. The three-dimensional surface topography of specimens was measured and characterized with a profile microscopy measuring system and scanning electron microscope. The friction coefficient showed a trend of decreasing first and then increasing with the increase in some surface topography parameters at lower pressure, such as the surface height arithmetic mean *Sa*, surface height distribution kurtosis *Sku*, surface volume average volume *Vvv*, and surface center area average void volume *Vvc*, which are the ISO 25178 international standard parameters. The effects of surface topographic parameters on friction were analyzed and the wear mechanism of the worn surface was presented. The wear characteristics of the samples were mainly characterized as strain fatigue, grinding, and scraping. The results provide a theoretical basis for the functional characterization of surface topography.

## 1. Introduction

Interface phenomena play a decisive role in the application of engineering parts, and their assessment and control are necessary in the development of many advanced fields such as electronics, information technology, energy, optics, and tribology [1,2]. Surface topography, as a basic feature of solid surface structures in the field of interface phenomena, has a decisive influence on friction, wear, and lubrication under mixed lubrication and dry friction conditions. The surface topography is a random structure composed of microscopic peaks and valleys formed during the manufacturing process and is widely present in various engineering practices. The surface topography not only contains manufacturing information, but also has a quality assurance effect, and the microscopic random structure of the surface topography has an important influence on many properties such as sealing, friction, interfacial heat transfer, and information storage. Sayles [3] analyzed the relationship between surface topography and functional properties and emphasized the potential importance of improving the functional properties of the friction of mechanical parts through surface topography. Mathia and Pawlus gave examples of the effects of various surface topographies and pointed out the need for surface characterization and testing when studying the effects of surface topography on tribological properties [4]. Bruzzone et al. believed that studying the relationship between surface topography, function, and application is a very complicated task, with particular emphasis on tribology [5]. The research by Gachot and Costa [6] pointed out that the dependence of surface topography on the mechanism of how it affects lubrication is not fully understood. It is still necessary to introduce different theories and methods to study this question. The acquisition of specific functional properties depends on the control of the matching surface microstructure and the complexity and diversity of the surface topography microstructure [7], such as the self-similarity, multi-scale characteristics, singularity, and some features that have not yet been recognized and understood. It also makes the study of the relationship between surface topography and functional properties a challenging research topic. Therefore, studying the complex characteristics of microscopic surface topography, proposing appropriate characterization methods for specific complex characteristics, and comprehensively clarifying the influence mechanism of the surface topography microstructures on friction or lubrication are of great significance to improve the quality of products and the performance of tribological systems.

Traditional surface topography is mainly concerned with the microscopic geometry of the surface. More attention is paid to the height information of the surface. Its function is to control the manufacturing precision of the surface. As for the actual performance of the surface, it cannot be characterized. The characterization and description of the surface topography structure are mainly based on the method of mathematical statistics [8]. Their main contribution is to make people have a deep understanding of the amplitude characteristics of surface topography. Early research mainly studied the influence of parameters, such as surface arithmetic mean on friction and wear characteristics [9]. Recent studies have also expanded the effects of surface skewness and kurtosis [10]. These studies have played a positive role in the surface topography properties.

The traditional characterization methods of three-dimensional surface topography include the reference parameter method, motif method, fractal method, watershed method, and wavelet method. These methods have been extended from two-dimensional surface topography characterization parameters evolving to three-dimensional surface topography characterization methods, which have made great breakthroughs in describing the spatial characteristics of surface topography. Pradeep et al. [11,12,13] studied the relationship between the friction coefficient and partial roughness parameters and found that the friction coefficient did not change significantly with the change in Ra, and the average slope of the contour had the greatest influence on the average friction coefficient. Many scholars have found that surfaces with the same surface roughness height but different surface skewness could lead to a great difference in tribological properties [14]. Hao et al. combined wavelet analysis and the watershed method, proposed feature-based surface characterization parameters, and studied their effect on friction characteristics [15]. Lou et al. studied the surface topography characterization problem based on the motif and pointed out the complementary effect of this method and the characterization based on midline surface topography [16].

The above work has played a positive role in studying the spatial structure characteristics of surface topography on friction, but it also has its limitations and deficiencies. One of them is the existence of too many parameters to characterize the three-dimensional surface, and there is a high correlation between them. There has been no uniform standard for the selection and processing of parameters by the researchers. In response to this problem, the International Organization for Standardization (ISO) 25178 surface topography characterization standard introduced in 2007 can be said to be a milestone in the development of surface characterization. It is based on the characterization of statistical methods and incorporates the advantages of other characterization methods. One of the main contributions of it is to replace hundreds of surface topographical parameters with 30 relatively independent parameters, and to establish an international standard for surface topography description methods.

Aiming at the need of surface topography research in the metal drawing process, Pfestorf and Engel proposed a mechanical rheological model to characterize the surface topography by studying the bearing form between interfaces, and established the closed void volume and contact volume of the contact interface [17]. The algorithm, for the first time, quantified the valleys of rough surfaces [18,19,20]. They proposed the concept of open empty body area and closed empty body area. The former causes the lubricant to be squeezed under external load to generate hydrodynamic pressure, and the latter causes the lubricant to flow under the action of external load to form hydrostatic pressure. The corresponding parameters show the ability of the surface to store and transport lubricants, all of which have a significant impact on the friction and lubrication properties of the surface. In the contact interface, peaks and ridges act as points of high contact stress and wear and produce abrasive particles and debris. In addition, pits and valleys will affect the characteristics of lubrication and fluid storage [20]. However, according to current data, research in this area is still in its infancy because it is difficult to parameterize the functional properties of three-dimensional surface topography. Etsion also pointed out that the optimization of surface topography parameters is still a theoretical problem to be solved when studying the application of laser surface textures [21]. In addition, the problem of incomplete information during processing such as the peak-to-valley boundary [22] is a problem that has not been satisfactorily solved.

Although many scholars have gradually established a link between one or a small part of the surface topography parameters and interface functions, a lot of experimentation is still needed to explore the relationship between them. This is because there are still many surface topography parameters that have uncertain effects on the functional properties and different effects on different material interfaces. In addition, designing and manufacturing a surface that meets the requirements of use is the ultimate goal. It relies on the research basis for establishing the relationship between surface topography parameters and interface characteristics. The purpose of this paper is to reveal the influence of surface topography parameters on the tribological properties of the contact interface. Four parameters in the ISO 25178 standard are selected to explore the principle between these parameters and the friction coefficient of the aluminum alloy contact interface. The research results can provide a basis method for the further study of the influence of other parameters on the tribological properties of the interface.

## 2. Materials and Methods

The 6000 series aluminum alloy is widely used in the automotive industry. The heat-treated 6000 series aluminum alloy sheet can be applied to the automobile body frame. European car manufacturers mostly use aluminum 6016 because of its good performance. After heat treatment at 180 °C for 30 min, the 6016 aluminum alloy precipitates a large amount of β phase from the supersaturated solid solution, and this needle-like precipitate can cause the alloy to exhibit a remarkable strengthening/hardening effect. Therefore, this experiment used two relatively hard materials as the pair of grinding parts, in order to explore the influence of surface topography parameters on friction and wear, and further determine whether aluminum 6016 is a material suitable as a vehicle chassis material. The upper test piece material was aluminum alloy 6016, and its tensile strength σ_b_ was 235 MPa. The plastic material had no obvious yield limit, and the yield index σ_0.2_ was 125 MPa. Aluminum alloy 6016 was processed into cylindrical pins with a contact surface diameter of 8 mm and a length of 28 mm, which is shown in Figure 1a. The counterpart was chosen as CoCrMo because of its hardness. Co powder is a type of Stellite self-melt alloy, with some chemical elements such as Cr, Si, and so on, as shown in Table 1. It was processed into a 60 mm × 52 mm × 8 mm thin plate, which is shown in Figure 1b. The smoother the wear surface, the lower the coefficient of friction and the lower the wear. Therefore, this experiment used 320# and 800# metallographic sandpaper for grinding the CoCrMo surface of the test piece to a rough surface of *Ra* = 0.4 μm, which minimizes the influence on the wear result. Each new test was replaced with a new pair of grinding parts. Therefore, the surface parameters were slightly different. After testing by the authors, the error of the friction coefficient was within 10%. Aluminum alloy 6016 test surfaces were grinded into different roughness surfaces by different sandpaper sizes, and the surface height arithmetic mean parameters *Sa* were 0.902, 1.757, 2.866, 3.220, and 3.802.

As non-contact measurement does not damage the surface to be tested, this experiment used a non-contact measurement method to measure the aluminum alloy 6016 surfaces. The 3D profilometer used in the experiment was the VR series 3D profilometer VR-5000 produced by Keyence (Osaka, Japan). The measuring range was 206 mm × 104 mm, the measuring accuracy was 0.5 μm, and the imaging component was a 1 inch and 4-megapixel monochrome Complementary Metal Oxide Semiconductor (CMOS) camera (1 inch and 4 million pixels, Keyence, Osaka, Japan), as is shown in Figure 3a.

The surface height arithmetic mean *Sa* is the arithmetic mean of the height amplitudes of the points in the surface sampling range. It is one of the important parameters describing the height direction of the rough surface.
(1)$$Sa=\frac{1}{lxly}\int_{0}^{lx}\int_{0}^{ly}\left|z\left(x,y\right)\right|dxdy\approx \frac{1}{MN}\sum_{i=1}^{M}\sum_{j=1}^{N}\left|z\left(x_{i},y_{i}\right)\right|,$$
where *z* (*x*, *y*) is the height amplitude of each sampling point on the surface, *lx* and *ly* are the boundary lengths of the sampling area, and *M* and *N* are the sampling points along the *X* direction and *Y* direction, respectively.

The surface height distribution kurtosis *Sku* is a measure of the severity of the change in the surface topography height distribution curve. It is defined as
(2)$$Sku=\frac{1}{Sq^{4}}\int_{-\infty }^{+\infty }\int_{-\infty }^{+\infty }z^{4}\left(x,y\right)p\left(z\right)dxdy\approx \frac{1}{MNSq^{4}}\sum_{i=1}^{M}\sum_{j=1}^{N}z^{4}\left(x,y\right),$$
where *Sq* denotes the sampling area and *p* (*z*) denotes the probability density function of the height amplitude distribution function *z* (*x*, *y*) of each sampling point in the sampling area. For a symmetrical surface Gaussian surface, the surface height distribution kurtosis value is equal to three. If the surface height distribution kurtosis value is greater than 3, it indicates that the surface height distribution changes more gently, and the surface points are concentrated near the reference plane. If the surface height distribution kurtosis value is less than 3, it indicates that the surface height distribution changes more sharply, and the surface has a higher peak or a lower trough.

*Vvv* and *Vvc* are all functional parameters, where *Vvv* is the average void volume in the surface valley region and *Vvc* is the average void volume in the central region of the surface, representing the average empty body when the surface bearing capacity is 80%–100% and 10%–80%, respectively. Void volume, used to describe the microstructural properties of the surface topography, characterizes the surface’s oil storage and lubrication capabilities.

Table 2 shows the characterization parameters of the three-dimensional surface topography of the five samples before the test. Figure 2 shows the 3D profiles and two-dimensional (2D) roughness curves of the surface of the five samples and the CoCrMo counterpart.

The experiment used the multi-functional friction and wear testing machine produced by Rtec Instrument Co., Ltd., with a maximum load of 5000 N, and the “X” and “Y” direction bidirectional motor control system, as is shown in Figure 3b. The test lubricant was selected from 32# mechanical oil, and the measured kinematic viscosity at 40 °C was 33.5 mm/s. The test was carried out at room temperature, which was about 20 °C, and the tensile direction of the test piece was perpendicular to the direction. As it was the first trial, the load was chosen in a wide range to measure the relationship between the surface topography parameters and the friction coefficient. The sliding speed of the test piece was set to 0.5 mm/s, and the contact pressure was 27.22, 54.44, 81.67, 108.89, and 136.11 MPa. The values of the friction coefficients of the test pieces I, II, III, IV, and V under different contact pressures were obtained by the average of three experiments.

The worn morphologies of specimens were observed by SEM (JEOL JSM-IT300, Tokyo, Japan) in order to study the wear mechanism. Every experiment lasted half an hour and the samples were ultrasonically cleaned in isopropyl alcohol for 10 min.

## 3. Results and Discussion

Based on the surface morphology and friction test results, the influence of the three-dimensional surface topography parameters on the friction characteristics was discussed. Figure 4 shows the relationship between the three-dimensional characterization parameters *Sa*, *Sku*, *Vvv*, and *Vvc* and the friction coefficient at a sliding speed of 0.5 mm/s. It can be seen from Figure 3 that with the increase in the height parameter *Sa*, the functional parameters, *Sku*, *Vvv*, and *Vvc*, and the friction coefficient first decrease and then increase, and the change in the friction coefficient produces a critical point, which is related to the critical characteristics of the surface topography [23,24]. In addition, as the contact pressure increases, some of the friction coefficient gradually decreases.

*Sa* characterizes the surface roughness state, *Sku* characterizes the height difference of the surface, and *Vvv* and *Vvc* characterize the void volume and oil storage capacity, respectively, and are related to the empty body area occupancy rate in this test, while the empty body area occupancy rate determines the rough contact interface. Under the sliding test between the test piece and the counterpart, the surface of the test piece tends to be flat; therefore, when the four parameters related to the area occupancy of the empty body are small, the actual contact area is large and the friction coefficient is large. When the four parameters are further increased, the actual contact area is reduced and the empty body area occupancy rate is gradually increased. The appropriate empty body volume can produce a better dynamic pressure lubrication effect, and the friction coefficient will decrease. When the empty body area occupancy rate is too large, the actual contact area is too small, the Hertz contact stress will cause plastic deformation of the sample material, the dynamic pressure lubrication effect is significantly reduced, and the friction coefficient will increase as the Hertz contact pressure increases. Therefore, it can be concluded that the friction coefficient of the rough contact interface is related to the area occupancy of the void body and some three-dimensional characterization parameters that could describe the anti-friction effect, which lays the theoretical foundation for the optimal design and manufacture of the surface topography basis.

Figure 4 also reveals that as the relative slip velocity is constant, the friction coefficient decreases with increasing contact load, and the friction coefficient changes with the load. The friction coefficient can be expressed as *μ* = *SA/P*, where *μ* refers to the friction coefficient, *A* is the apparent contact area, *S* is the shearing stress, and *P* is the applied load; thus, *P/A* is the apparent pressure, while *S* remains constant, and, therefore, the friction coefficient of the material decreases with the increase in normal load [25].

As the occupancy of the void area increases, the friction coefficient changes non-monotonically. Jensen introduced the network method into the contact interface problem. The results show that there is a critical feature in the contact problem that takes into account the characteristics of the surface peak and valley [26]. This critical property can be described by a percolation model and is closely related to the surface topography. For example, Bottiglione et al. [27] used percolation theory and contact mechanics theory to propose that when the surface topography and contact force reach a percolation threshold in a sealing system, a percolation channel is formed between the contact interfaces to cause fluid leakage. It reveals how the statistical characteristics of the interface in the system, the applied load, and the sealing geometry affect the leakage. Dapp et al. [28] studied the fluid flow of elastic solid contact on a random roughness self-imitation surface. Numerical simulations show that the elastic deformation reduces the contact area and improves the percolation effect compared to conventional sealing methods. When studying the surface leakage problem, Persson et al. pointed out that the critical phenomenon induced by the rough surface is a problem that has not yet been clarified and urgently studied [29].

The five samples under the pressure of 27.22 MPa were selected for microstructure characterization. SEM morphologies of the worn surfaces of the specimens at that testing condition are shown in Figure 5. It can be seen from the figures that some typical characteristics can be observed on the worn surface, such as ploughing-cutting, strain fatigue, abrasion, scratching, and so on. It can be seen from Figure 4 that as the void volume of the surface increases, the wear becomes lighter and lighter and the wear surface hardly sees the wear scar, as shown in Figure 5e. As the surface height arithmetic mean Sa increases, it experiences a large contact stress during wear, causing surface peaks to be worn away, resulting in strain fatigue and grinding, as shown in Figure 5a. When the surface height distribution kurtosis *Sku* is large, the surface is relatively flat, the Hertz stress is small, and the wear is light, as shown in Figure 5c,d. To sum up, when the void volume and height distribution kurtosis of the rough random surface are larger, the performance of the storing lubricating oil and lubrication is better, and the wear is less during the sliding process. When the height arithmetic mean of the rough random surface is larger, the contact pressure is increased, the wear is more severe, and deep wear marks and plastic deformation occur. It is known from the experimental results that the surface topographical parameters have a great influence on the wear behavior of the rough contact interface.

## 4. Conclusions

In order to obtain the effect of surface topography parameters on the friction of a rough contact interface, an experiment was carried out based on fluid dynamic pressure conditions.

The influence of surface topography characterization parameters on the friction of a rough contact interface has been discussed in many research works. Four ISO 25178 surface topography international standard characteristic parameters were selected in this paper: The arithmetic mean of the height amplitudes of the points in the surface sampling range, the surface height distribution kurtosis *Sku*, the surface volume average volume *Vvv*, and the surface center area average void volume *Vvc*. The variation in the friction coefficient of the rough contact interface with the surface topography parameters was obtained. The results of friction experiments have shown that with the increase in the three-dimensional characterization parameters *Sa*, *Sku*, *Vvv,* and *Vvc*, the friction coefficient showed a trend of decreasing first and then increasing, which is related to the critical characteristics of surface topography. When the surface topography parameter was small, the fluid dynamic pressure lubrication effect was good and the friction coefficient was low. When the surface topography parameters were large, the Hertzian stress caused the surface to be elastically deformed and the friction coefficient increased. Through the observation of the wear surface, it could be seen that the void volume parameters of the rough surface determined the lubrication performance of the contact interface. The larger the void volume parameter of the surface, the easier the lubricant can penetrate, and the smaller the surface wear. The main wear characteristic of the surface was ploughing-cutting, strain fatigue, and abrasion.

The results could be used to supplement the existing research on hydrodynamic lubrication and surface topography characterization.

## Figures and Tables

**Figure 1 materials-12-02762-f001:**
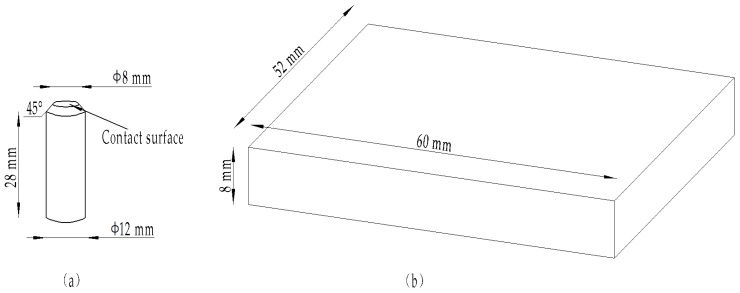
Experiment samples for (**a**) pin of aluminum alloy 6016 and (**b**) plate of CoCrMo.

**Figure 2 materials-12-02762-f002:**
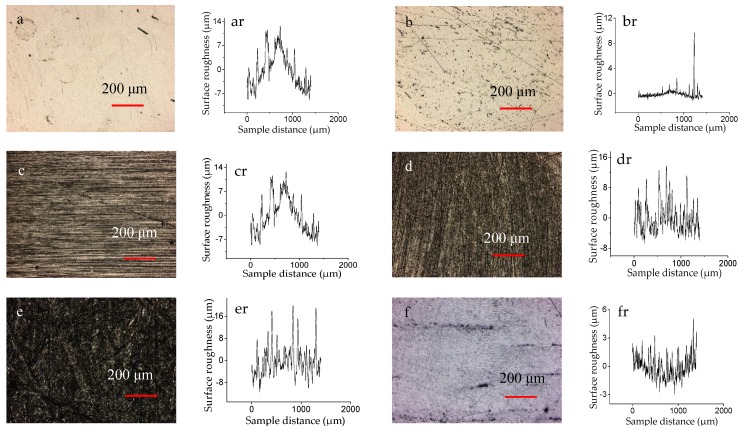
The 3D profiles and 2D roughness curves of the surface of the samples: ((**a**) and (ar)) for I, ((**b**) and (br)) for II, ((**c**) and (cr)) for III, ((**d**) and (dr)) for IV, ((**e**) and (er)) for V, and ((**f**) and (fr)) for counterpart.

**Figure 3 materials-12-02762-f003:**
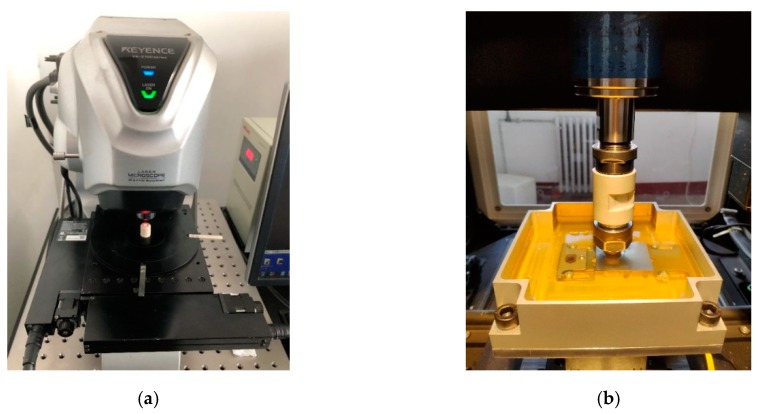
Diagram of testing machine (**a**) Surface topography measurement system (VR-5000, Keyence, Osaka, Japan); (**b**) friction testing machine (Multi Function Tribometer MFT-5000, Rtec Instrument Co., Ltd., San Jose, CA, USA).

**Figure 4 materials-12-02762-f004:**
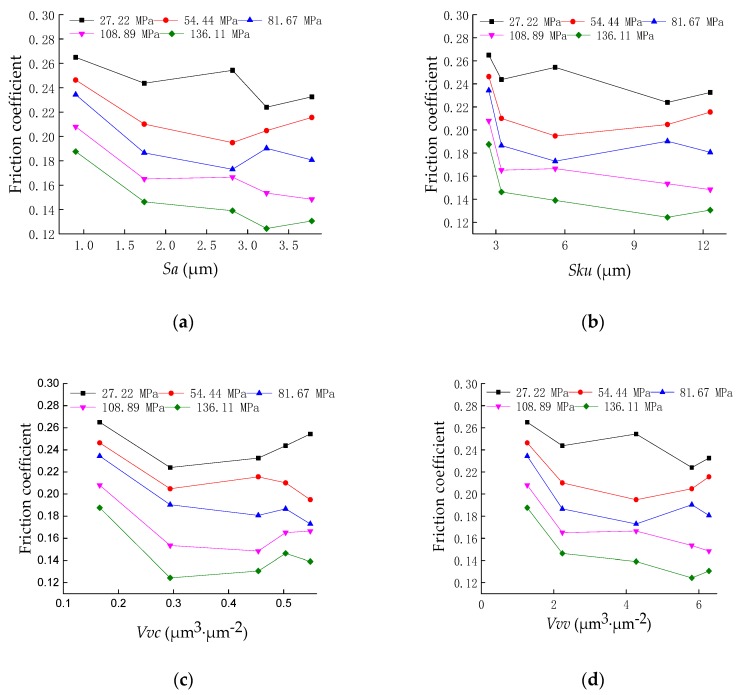
Friction coefficient as a function of 3D characterization parameters. (**a**) Friction coefficient as a function of *Sa*; (**b**) friction coefficient as a function of *Sku*; (**c**) friction coefficient as a function of *Vvv*; (**d**) friction coefficient as a function of *Vvc*.

**Figure 5 materials-12-02762-f005:**
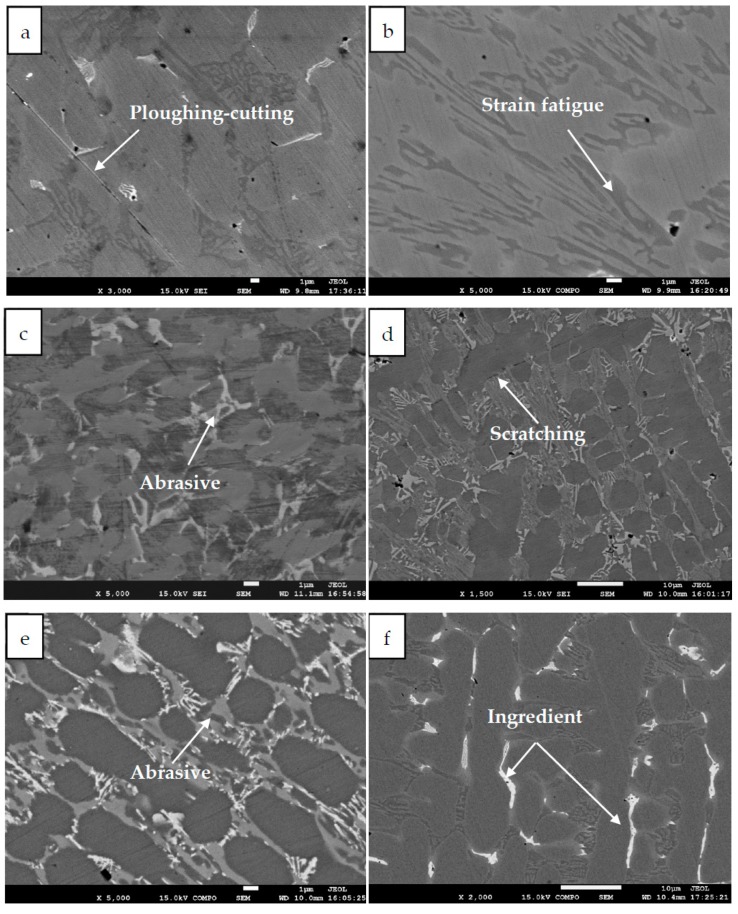
SEM morphologies of worn surfaces: (**a**) I, (**b**) II, (**c**) III, (**d**) IV, (**e**) V, and (**f**) counterpart under fluid dynamic lubrication condition at the slide speed of 0.5 mm/s and pressure of 27.22 MPa.

**Table 1 materials-12-02762-t001:** Chemical composition (wt. %) of the tested Co powder.

C	Cr	Si	W	Fe	Mo	Ni	Co	Mn
1.40	29.50	1.45	8.25	3.00	1.00	3.00	Balance	1.00

**Table 2 materials-12-02762-t002:** The 3D surface parameters before testing.

3D Surface Parameters	Five Samples					Counter Part
	I	II	III	IV	V	CoCrMo
*Sa*/μm	0.902	1.757	2.866	3.220	3.802	0.392
*Sku*/μm	2.695	3.046	5.646	10.186	10.734	3.056
*Vvv*/(μm^3^·μm^−2^)	0.1661	0.5323	0.5041	0.2912	0.4845	0.093
*Vvc*/(μm^3^·μm^−2^)	1.2698	2.2575	4.3094	5.6450	6.2281	0.989

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
