# Peer review of "Effect of Surface Topography Parameters on Friction and Wear of Random Rough Surface"

_materials, 2019, doi:10.3390/ma12172762_

Round 1
Reviewer 1 Report
The paper deals with the basic issue relating to influence of surface preparation on wear results. The authors studied four roughness ISO parameters in relation to wear test results.
General comments:
1. The goal of the study is not presented sufficiently. Please emphasize the novelty of your work in relation to current state of art. The main findings should be stressed in the conclusions section.
2. Please justify why the Al alloy and Co-alloy pair was studied.
3. Please explain in the paper if the roughness of the counter part influences on the wear result?
4. The title of the manuscript includes "wear". Thus, the wear mechanism should be studied toughly. I suggest to use higher magnification SEM photos and add SEM for the counter part.
Minor remarks:
5. Please add in the paper the chemical composition of Co-based alloy.
6. Please write what was the roughness of the counterpart. Add the 4 investigated parameters for that counter part sample (only 2D parameter (Ra) is given.) I suggest you to add that parameters in table 1.
7. Did you use the same (only one) CoCrMo sample?
8. Fig. 1: the value of chamfering in (a) should be marked, and 8 and 12 mm should be described with diameter sign Ø. Also (b) dawning should be presented with appropriate proportions (now, the dimension of 52 is relatively to small in comparison to 60mm).
9. Fig. 4 is not clear. Please provide SEM photos with higher magnification. Now it is impossible to see and describe the wear traces.
10. L220-222: last phrase is too general.
11. In conclusion: add the main finding - explain briefly why the friction firstly increases and subsequently decreases.
12. L236: "As the contact pressure increases, some of the coefficient of friction decreases" - phrase is not clear.
Author Response
According to the reviewers' comments, all revisions in manuscript have been clearly highlighted using the "red font" in Microsoft Word.
The goal of the study and the novelty of the work have been presented in the abstract and L102-L108. And the main findings have been stressed in the conclusions section which are in L257-263. The reason why the counterparts were chosen has been added in L114-122. The roughness of the counterpart influences on the wear result slightly because the hardness of the CoCrMo is bigger than Al a lot. The explanation is in L128-131. The higher magnification SEM photos and counterpart SEM photos have been added in Fig. 5. The chemical composition of Co-based alloy has been added inTable.1. The surface parameters of the counterpart were given in Table. 2. Each new test is replaced with a new pair of grinding parts, as shown in L131. The fig.1 has been rectified. The higher magnification SEM photos and counterpart SEM photos have been added in Fig. 5. This paragraph has been removed because of its inaccurate. The reasons have been added in L267-270. This sentence has been removed by accurate expressions in L257-263.Reviewer 2 Report
This paper presents the results and discussion obtained after conducted experimental study of the influence of some 3D-roughness parameters over tribological characteristics for aluminium alloy 6016 - CoCrMo sliding pairs, under fluid dynamic pressure lubrication condition. A detailed overview of existing tribological and wear mechanisms concepts is provided. Authors are selected four ISO 25178 surface topography parameters (such as the surface height arithmetic mean Sa, the surface height distribution kurtosis Sku, the surface volume average volume Vvv, and the surface center area average void volume Vvc), to investigate their effect over friction coefficient in five different load conditions and the wear mechanism of the sliding pair. The results obtained and the analyzes drawn are very similar to other conducted studies for other combinations of materials in friction pairs. For the main contribution of this work, I consider the use of a 3D profilometer to determine the three-dimensional parameters of the roughness and a scanning electron microscope to obtain the morphological images of the worn surface of the specimens.
The information in the article is properly arranged and the requirements of the Materials journal as generally have been met.
Overall I recommend acceptance but only after the following questions and concerns are addressed.
- Although the abstract states that the results obtained in the paper provide a "theoretical basis" for further studies of the problem, the theoretical element is not sufficiently clarified in the content of the article. It presents only analyses and conclusions, based on the results obtained for a specific experimental set-up, consist of two specific materials, specimens with certain shape and dimensions, friction with specific lubricate conditions, etc.
- It is not well explained why the experimental specimens in the sliding pair were selected to be aluminium 6016 and CoCrMo alloy? Are there any examples of such real sliding pairs with some practical application in the pointed areas of interest in the study? This leads to questions such as why did the authors choose these values of contact pressures and sliding speed?
- The processing methods used for obtain both of experimental friction pairs parts, are not clarified enough. For the aluminium pins (I guess) the authors says that 320# and 800# metallographic sandpaper grinds were used, but what finishing process was used for the CoCrMo plate? What is the initial roughness of the CoCrMo-plate? Are one and same CoCrMo-plate was used for all experimental trials? How its roughness change during the overall experimental study?
I have also some minor remarks and recommendations:
- Why in the section "Materials and Methods" the formulas for roughness parameters Sa and Sku are given, but for the parameters Vvv and Vvc are not?
- The diameter symbols prefixes before the diameters dimensions of the pins, shown on Fig. 1 should be added.
- It is recommended that some images from the 3D profilometer, or 2D cross sections of the roughness profile from them, be added to the results to support the values given in Table 1 for the roughness parameters.
Author Response
According to the reviewers' comments, all revisions in manuscript have been clearly highlighted using the "red font" in Microsoft Word.
The explanation of the goal of the study and the novelty of the work have been presented in the abstract and L102-L108. The reason why the counterparts were chosen has been added in L114-122. The processing methods have been clarified in L126-135 and Table. 1 and 2. Because the functional parameters Vvv and Vvc are obtained by the surface bearing curve, which are not calculated, as shown as http://en.wikipedia.org/wiki/ISO_25178. The Fig.1 has been rectified. The 3D photos and 2D roughness profiles have been added as Fig.2.Round 2
Reviewer 2 Report
I accept all revisions in the paper that authors have made.
My recommendation is the revised work to be accepted for publishing.